# SegViT: Semantic Segmentation with Plain Vision Transformers

**Bowen Zhang**[1*], **Zhi Tian**[2*], **Quan Tang**[4], **Xiangxiang Chu**[2],
**Xiaolin Wei**[2], **Chunhua Shen**[3], **Yifan Liu**[1]

[1] The University of Adelaide, Australia     [2] Meituan Inc.
[3] Zhejiang University, China     [4] South China University of Technology, China

## Abstract

We explore the capability of plain Vision Transformers (ViTs) for semantic segmentation and propose the SegViT. Previous ViT-based segmentation networks usually learn a pixel-level representation from the output of the ViT. Differently, we make use of the fundamental component—attention mechanism, to generate masks for semantic segmentation. Specifically, we propose the Attention-to-Mask (ATM) module, in which the similarity maps between a set of learnable class tokens and the spatial feature maps are transferred to the segmentation masks. Experiments show that our proposed SegViT using the ATM module outperforms its counterparts using the plain ViT backbone on the ADE20K dataset and achieves new state-of-the-art performance on COCO-Stuff-10K and PASCAL-Context datasets. Furthermore, to reduce the computational cost of the ViT backbone, we propose query-based down-sampling (QD) and query-based up-sampling (QU) to build a *Shrunk* structure. With the proposed *Shrunk* structure, the model can save up to $40\%$ computations while maintaining competitive performance.

## 1 Introduction

Semantic segmentation is a dense prediction task in computer vision that requires pixel-level classification of an input image. Fully Convolutional Networks (FCN) [1] are widely used in recent state-of-the-art methods. This paradigm includes a deep convolutional neural network as the encoder/backbone and a segmentation-oriented decoder to provide dense predictions. A $1\times1$ convolutional layer is usually applied to a representative feature map to obtain the pixel level predictions. To achieve higher performance, previous works [2–4] focus on enriching the context information or fusing multi-scale information. However, the correlations among spatial locations are hard to model explicitly in FCNs due to the limited receptive field.

Recently, Vision Transformers (ViT) [5], which make use of the spatial attention mechanism are introduced to the field of computer vision. Unlike typical convolution-based backbones, the ViT has a plain and non-hierarchical architecture that keeps the resolution of the feature maps all the way through. The lack of the down-sampling process (excluding tokenizing the image) brings differences to the architecture to do the semantic segmentation task using ViT backbone. Various semantic segmentation methods [6–8] based on ViT backbones have achieved promising performance due to the powerful representation learned from the pre-trained backbones. However, the potential of the attention mechanism is not fully explored.

Different from previous per-pixel classification paradigm [6–8], we consider learning a meaningful class token and then finding local patches with higher similarity to it. To achieve this goal, we propose

---

*The first two authors contributed equally. CS is the corresponding author, e-mail: `chunhua@me.com`

36th Conference on Neural Information Processing Systems (NeurIPS 2022).

the Attention-to-Mask (ATM) module. More specifically, we employ a transformer block that takes the learnable class tokens as queries and transfers the spatial feature maps as keys and values. A dot-product operator calculates the similarity maps between queries and keys. We encourage regions belonging to the same category to generate larger similarity values for the corresponding category (*i.e.* a specific class token). Fig. 1 visualizes the similarity maps between the features and the 'Table' and 'Chair' tokens. By simply applying a `Sigmoid` operation, we can transfer the similarity maps to the masks. Meanwhile, following the design of a typical transformer block, a `Softmax` operation is also applied to the similarity maps to get the cross attention maps. The 'Table' and 'Chair' tokens are then updated as in any regular transformer decoders, by a weighted sum of the values with the cross attention maps as the weights. Since the mask is a byproduct of the regular attentive calculations, negligible computation is involved during the operation.

Building upon this efficient ATM module, we propose a new semantic segmentation paradigm with the plain ViT structure, dubbed SegViT. In the paradigm, several ATM modules are employed on different layers, and we get the final segmentation mask by adding the outputs from different layers together. SegViT outperforms its ViT-based counterparts with less computational cost. However, compared with previous encoder-decoder structures that use hierarchical networks as encoders, ViT backbones as encoders are generally heavier. To further reduce the computational cost, we employ a *Shrunk* structure consisting of query-based down-sampling (QD) and query-based up-sampling (QU). The QD can be inserted into the ViT backbone to reduce the resolution by half and QU is used parallel to the backbone to recover the resolution. The *Shrunk* structure together with the ATM module as the decoder can reduce up to $40\%$ computations while maintaining competitive performance.

We summarize our main contributions as follows:

- We propose an Attention-to-Mask (ATM) decoder module that is effective and efficient for semantic segmentation. For the first time, we utilize the spatial information in attention maps to generate mask predictions for each category, which can work as a new paradigm for semantic segmentation.

- We managed to apply our ATM decoder module to the plain, non-hierarchical ViT backbones in a cascade manner and designed a structure namely SegViT that achieves mIoU $55.2\%$ on the competitive ADE20K dataset which is the best and lightest among methods that use ViT backbones. We also benchmark our method on the PASCAL-Context dataset ($65.3\%$ mIoU) and COCO-Stuff-10K dataset ($50.3\%$ mIoU) and achieve new state-of-the-art performance.

- We further explore the architecture of ViT backbones and work out a *Shrunk* structure to apply to the backbone to reduce the overall computational cost while still maintaining competitive performance. This alleviates the disadvantage of ViT backbones that are usually more computationally intensive compared to their hierarchical counterparts. Our *Shrunk* version of SegViT on the ADE20K dataset reaches mIoU $55.1\%$ with the computational cost of 373.5 GFLOPs which is about $40\%$ off compared to the original SegViT (637.9 GFLOPs).

## 2   Related Work

**Semantic segmentation.**   Semantic segmentation which requires pixel-level classification on an input image is a fundamental task in computer vision. Fully Convolutional Networks (FCN) used to be the dominant approach to this task. Initial per-pixel approaches such as [9, 10] attribute the class label to each pixel based on the per-pixel probability. To enlarge the receptive field, several approaches [11, 12] have proposed dilated convolutions or apply spatial pyramid pooling to capture contextual information of multiple scales. With the introduction of attention mechanisms, [13, 14, 6] replace the feature merge conducted by convolutions and pooling with attention to better capture long-range dependencies.

Recent works [15, 8, 16] decouple the per-pixel classification process. They reconstruct the structure by using a fixed number of learnable tokens and use them as weights for the transformation to apply on feature maps. Binary matching rather than cross-entropy is used to allow overlaps between feature maps and learnable tokens are used to dynamically generate classification probabilities. This paradigm enables the classification process to be conducted globally and alleviates the burden for the decoder to do per-pixel classification, which as a result, is more precise and the performance is

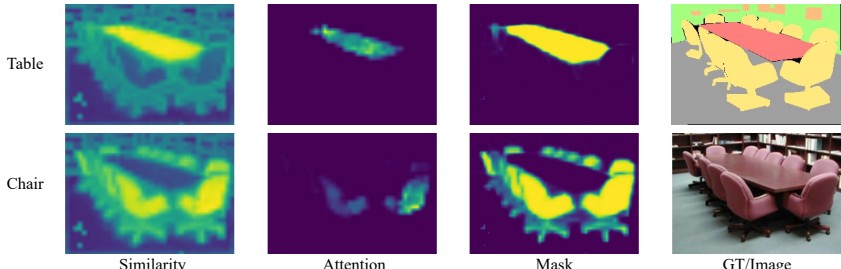

Table           Chair

Similarity        Attention        Mask        GT/Image

**Figure 1: The overall concept of our Attention-to-Mask decoder.** In a typical attentive process, the dot-product is first calculated between queries and keys to measure the similarity (as illustrated on the left). If the similarity map is applied with `Softmax` operation on the spatial dimension, the output is the typical attention map (multiple heads are summed together). However, if the same similarity map is applied with a per-pixel operation `Sigmoid`, it produces a mask that indicates the area with certain similarity. Based on the assumption that the tokens within the same category have higher similarity, we can train a token vector to have high similarity within tokens of the specific category and low similarity elsewhere. In the meantime, this process does not violate the attention mechanism. Thus, it can process alongside the original transformer layers.

generally better. However, for those methods, the feature map is still calculated in a static manner, usually requiring feature merge modules such as FPN [4].

**Transformers for vision.** Attention-based transformer backbones have become powerful alternatives to standard convolution based networks for image classification tasks. The original ViT [5] is a plain, non-hierarchical architecture. Various hierarchical transformers such as [17–21] have been presented afterwards. These methods inherit some designs from convolution based networks such as hierarchical structures, pooling and down-sampling with convolutions. As a result, they can be used as a straightforward replacement for convolutional based networks and applied with previous decoder heads for tasks such as semantic segmentation.

**Plain-backbone decoders.** High-resolution feature maps generated by backbones are important for dense prediction tasks such as semantic segmentation. Typical hierarchical transformers use feature merge techniques such as FPN [4] or dilated backbones to generate high-resolution feature maps. However, for plain, non-hierarchical transformer backbones, the resolution remains the same for all layers. SETR [6] proposed a simple strategy to treat transformer outputs in a sequence-to-sequence perspective to solve segmentation tasks. Segmenter [8] joints random initialized class embeddings and the transformer patch embeddings together and applies several self-attention layers to the joint token sequence to obtain updated class embeddings and patch embeddings semantic prediction. In our study, we consider learning a class token and then finding local patches with higher similarities with the help of the attention map, making the inference process more direct and efficient.

## 3 Method

### 3.1 Encoder

Given an input image $I \in \mathbb{R}^{H \times W \times 3}$, a plain vision transformer backbone reshapes it into a sequence of tokens $\mathcal{F}_0 \in \mathbb{R}^{L \times C}$ where $L = HW/P^2$, $P$ is the patch size and $C$ is the number of channels. Learnable position embeddings of the same size of $\mathcal{F}_0$ are added to capture the positional information. Then, the token sequence $\mathcal{F}_0$ is applied with $m$ transformer layers to get the output. We define the output tokens for each layer as $[\mathcal{F}_1, \mathcal{F}_2, \ldots, \mathcal{F}_m] \in \mathbb{R}^{L \times C}$. Typically, a transformer layer consists of a multi-head self-attention block followed by a point-wise multilayer perceptron block with layer norm in between and then a residual connection is added afterward. The transformer layers are stacked repetitively several times. For a plain vision transformer like ViT, there are no other modules involved and for each layer, the number of the tokens is not changed.

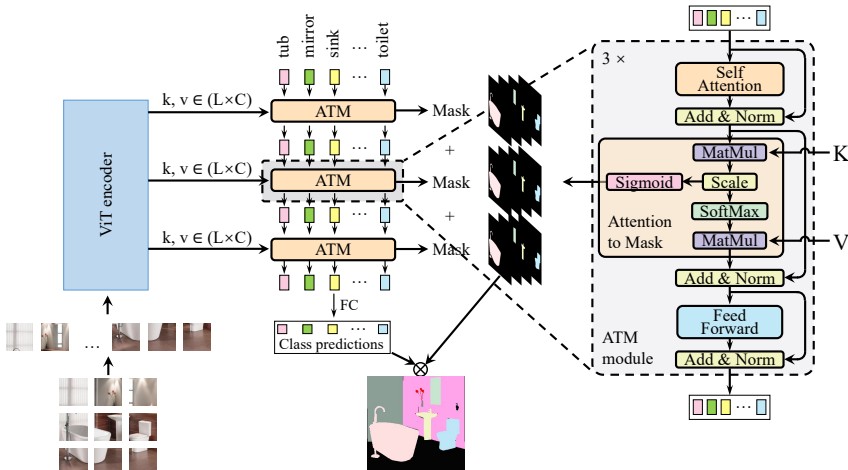

**Figure 2: The overall SegViT structure with the ATM module.** The Attention-to-Mask (ATM) module inherits the typical transformer decoder structure. It takes in randomly initialized class embeddings as queries and the feature maps from the ViT backbone to generate keys and values. The outputs of the ATM module are used as the input queries for the next layer. The ATM module is carried out sequentially with inputs from different layers of the backbone as keys and values in a cascade manner. A linear transform is then applied to the output of the ATM module to produce the class predictions for each token. The mask for the corresponding class is transferred from the similarities between queries and keys in the ATM module.

## 3.2 Decoder

**Mask-to-Attention (ATM).** Cross attention can be described as the mapping between two sequences of tokens. We define two token sequences as $\mathcal{G} \in \mathbb{R}^{N \times C}$ with the length $N$ equals to the number of classes and $\mathcal{F}_i \in \mathbb{R}^{L \times C}$. First, linear transformations are applied to each of them to form query (Q), key (K) and values (V), as presented by Eq. (1).

$$Q = \phi_q(\mathcal{G}) \in \mathbb{R}^{N \times C}, \ K = \phi_k(\mathcal{F}_i) \in \mathbb{R}^{L \times C}, \ V = \phi_v(\mathcal{F}_i) \in \mathbb{R}^{L \times C}, \tag{1}$$

The similarity map is calculated between the query and the key. Following the scaled dot-product attention mechanism, the similarity map and attention map are calculated by:

$$S(Q, K) = \frac{QK^T}{\sqrt{d_k}} \in \mathbb{R}^{N \times L},$$

$$Attention(\mathcal{G}, \mathcal{F}_i) = \texttt{Softmax}(S(Q, K))V \in \mathbb{R}^{N \times C}, \tag{2}$$

where $\sqrt{d_k}$ is a scaling factor with $d_k$ equals to the dimension of the keys. The shape of the similarity map $S(Q, K)$ is determined by the length of the two token sequences $N$ and $L$. The attention mechanism is then to update $\mathcal{G}$ by a weighted sum of $V$, where the weight assigned to the summation is the similarity map applied with `Softmax` along the dimension $L$.

Dot-product attention uses the `Softmax` function to exclusively concentrate the attention on the token that has the most similarity. However, we suppose that the tokens other than ones that yield maximum similarities are also meaningful. Based on this intuition, we design a lightweight module that generates semantic predictions more directly. To be more specific, we assign $\mathcal{G}$ as the class embeddings for the segmentation task and $\mathcal{F}_i$ as the output of layer $i$ of the ViT backbone. We pair a semantic mask to each token in $\mathcal{G}$ to represent the semantic prediction for each class. The calculation for the mask is:

$$Mask(\mathcal{G}, \mathcal{F}_i) = \texttt{Sigmoid}(S(Q, K)) \in \mathbb{R}^{N \times L} \tag{3}$$

The shape of the masks is $N \times L$, which can be further reshaped to $N \times H/P \times W/P$. The structure of the ATM mechanism is illustrated in the right part in Fig. 2. Masks are the middle output of the cross attention. The final output tokens from the ATM module are used for classification. We apply a linear transformation followed by a `Softmax` activation to the output class tokens to get class probability predictions. Note that we follow [15] to add a 'no object' category (Ø) in case the image

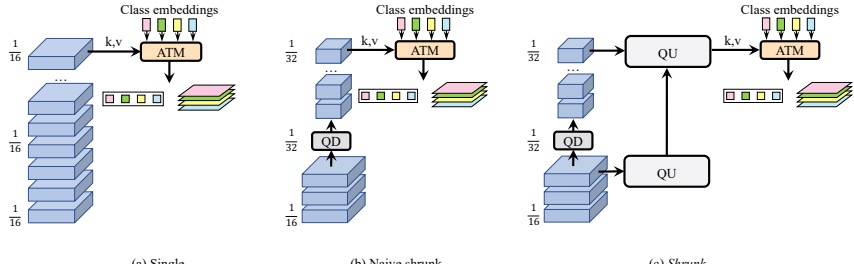

**Figure 3: The structure comparison between SegViT with a single layer and the *Shrunk* version.** (a) illustrates the SegViT structure with ATM module used once with the last layer of the ViT backbone as the input to generate predictions. (b) uses the query-based down-sampling (QD) module to implement a naive way to shrink the resolution of the features of the backbone from $1/16$ to $1/32$ and thus reduces the overall computational cost. (c) is the proposed (shrunk) version which applies the additional query-based up-sampling module. The *Shrunk* version can save up to 40% of computational cost when using the ViT-Large backbone without much sacrifice to the performance.

doesn't contain certain classes. During inference, the output is produced by the dot-product between the class probability and the mask groups.

Plain backbones such as ViT does not have multiple stages with features of different scale. Thus, structures such as FPN to merge features with multiple scales are not applicable. However, features other than the last layer contain rich low-level semantic information and are beneficial to the performance. We designed a structure that can make use of the feature maps from different layers of ViT to compact with our ATM decoder namely SegViT. In this study, we also found a way to compact the computational cost for the ViT backbone without sacrificing performance. This proposed *Shrunk* version of SegViT uses query-based down-sampling (QD) module together with a query-based up-sampling (QU) module to compress the ViT backbone and bring an overall reduction to the computational cost.

**The SegViT structure.** As illustrated in Fig. 2, an ATM decoder takes in $N$ tokens as the class embeddings and another sequence of tokens as the base to calculate keys and values for the ATM module to generate masks. The output of the ATM is $N$ updated tokens and $N$ masks corresponding to each class token. We use random initialized learnable tokens as the class embeddings and the output of the last layer of the ViT backbone as the base first. To make use of multi-layer information, the output of the first ATM decoder is then used as the class embeddings for the next ATM decoder with the output of another layer of the ViT backbone as the base. This process is repeated another time so that we can get three groups of tokens and masks. Formally, the loss function of each layer can be formulated as,

$$\mathcal{L}_{overall} = \mathcal{L}_{cls} + \mathcal{L}_{mask} = \mathcal{L}_{cls} + \lambda_{focal}\mathcal{L}_{IoU} + \lambda_{dice}\mathcal{L}_{dice} \tag{4}$$

In each group, the output tokens are supervised by the classification loss ($\mathcal{L}_{cls}$) which is mentioned above and the masks are summed orderly and supervised by the mask loss ($\mathcal{L}_{mask}$) which is a linear combination of a focal loss [22] and a dice loss [23] multiplied by hyper-parameters $\lambda_{focal}$ and $\lambda_{dice}$ respectively as in DETR [24]. The loss of all three groups are then summed together. We have further experiments to show that this design is beneficial and efficient.

**The *Shrunk* structure.** Plain transformer backbones such as ViT is known to have larger computational cost than their counterparts with similar performance. We propose a *Shrunk* structure using query-based down-sampling (QD) and up-sampling (QU). Since the shape of the output of the attention module is determined by the shape of the query, we can apply down-sampling before the query transformation to realize the QD or insert new query tokens during the cross attention to realize the QU. By changing the resolution with the number of query tokens, the spatial size is changed according to the cross attention, providing more flexibility to preserve (recover) important regions. To be more specific, in the QD layer, we use the nearest sampling to reduce the number of the query tokens while keep the size of the key and value tokens. When passing through a transformer layer, the values are weighted and summed by the attention map between query tokens and the key tokens. This is non-linear downsampling that will pay more attention to the important regions. In the QU

layer, we employ a transformer decoder structure [25] and initialize new learnable tokens as queries based on the desired output resolution.

As shown in Fig. 3, we design the SegViT structure with one single layer as the baseline (a). We first try a naive approach (b), which is to apply the QD once at the $1/3$ depth of the backbone (*e.g.*, the 8th layer of a backbone with 24 layers) to down-sample the resolution of the layer output from $1/16$ to $1/32$ so as to reduce the overall computational cost. The performance drops as expected since the QD process involves information lose.

To compensate for the information loss in the naive 'shrunk' version, we further apply two QU layers in parallel with the backbone. This is our proposed *Shrunk* version (c). The first QU layer takes in features with $1/16$ resolution from the low level of the backbone. Its output is then used as the query to make cross attention with the down-sampled features with $1/32$ resolution from the last layer of the backbone. The shape of the output of this QU structure is of $1/16$ resolution.

Directly reducing the number of the query tokens inevitably harms the final performance. However, with our designed QU layer and the ATM module, the *Shrunk* structure is able to reduce $40\%$ of overall computational cost while still being competitive in performance.

## 4 Experiments

### 4.1 Datasets

**ADE20K [26]** is a challenging scene parsing dataset which contains $20,210$ images as the training set and $2,000$ images as the validation set with 150 semantic classes.

**COCO-Stuff-10K [27]** is a scene parsing benchmark with $9,000$ training images and $1,000$ test images. Even though the dataset contains 182 categories, not all categories exist in the test split. We follow the implementation of mmsegmentation [28] with 171 categories to conduct the experiments.

**PASCAL-Context [29]** is a dataset with $4,996$ images in training set and $5,104$ images in the validation set. There are 60 semantic classes in total, including a class representing 'background'.

### 4.2 Implementation details

**Transformer backbone.** We use the naive ViT [5] as the backbone. In particular, we use its 'Base' variation for most ablation studies and provide results on the 'Large' variation. Since there can be a huge difference with different pre-trained weights, as suggested by Segmenter [8], we use the weights provided by Augreg [30] following the counterparts [8, 31] for a fair comparison. The weights are obtained by training on ImageNet-21k with strong data augmentation and regularization. For a simple reference, we report that for pre-trained weights provided by ViT [5] and Augreg [30], the mIoU scores using the same training recipe on ADE20K dataset are $51.7\%$ and $54.6\%$, respectively. **Training settings.** We use MMSegmentation [28] and follow the commonly used training settings. During training, we applied data augmentation sequentially via random horizontal flipping, random resize with the ration between $0.5$ and $2.0$ and random cropping ($512 \times 512$ for all except that we use $480 \times 480$ for PASCAL-Context and $640 \times 640$ for ViT-large on ADE20K). The batch size is 16 for all datasets with a total iteration of $160k$, $80k$ and $80k$ for ADE20k, COCO-Stuff-10k and PASCAL-Context respectively. **Evaluation metric.** We use the mean Intersection over Union (mIoU) as the metric to evaluate the performance. 'ss' means single-scale testing and 'ms' test time augmentation with multi-scaled $(0.5, 0.75, 1.0, 1.25, 1.5, 1.75)$ inputs. All reported mIoU scores are in a percentage format. All reported computational costs in GFLOPs are measured using the fvcore [2] library.

### 4.3 Comparisons with the State-of-the-art Methods

**Results on ADE20K.** Table 1 reports the comparison with the state-of-the-art methods on ADE20K validation set using ViT backbone. The SegViT uses the ATM module with multi-layer inputs from the original ViT backbone, while the *Shrunk* is the one that conducts QD to the ViT backbone and saves $40\%$ of the computational cost without sacrificing too much performance. Our method achieves

[2] https://github.com/facebookresearch/fvcore

**Table 1:** Experiment results on the ADE20K `val.` split. 'ms' means that mIoU is calculated using multi-scale inference. '†' means the models use the backbone weights pre-trained by AugReg [30]. '*' represents the model is reproduced under the same settings as the official repo. The GFLOPs is measured at single-scale inference with the given crop size.

| Method | Backbone | Crop Size | GFLOPs | mIoU (ss) | mIoU (ms) |
|---|---|---|---|---|---|
| UperNet* [32] | ViT-Base | $512 \times 512$ | >250 | 46.6 | 47.5 |
| DPT* [7] | ViT-Base | $512 \times 512$ | 219.8 | 47.2 | 47.9 |
| SETR-MLA* [6] | ViT-Base | $512 \times 512$ | 113.5 | 48.2 | 49.3 |
| Segmenter* [8] | ViT-Base | $512 \times 512$ | 129.6 | 49.0 | 50.0 |
| StructToken [31] | ViT-Base | $512 \times 512$ | >150 | 50.9 | 51.8 |
| SegViT (Ours) | ViT-Base | $512 \times 512$ | 120.9 | **51.3** | **53.0** |
| DPT* [7] | ViT-Large$^\dagger$ | $640 \times 640$ | 479.0 | 49.2 | 49.5 |
| UperNet* [32] | ViT-Large$^\dagger$ | $640 \times 640$ | >700 | 48.6 | 50.0 |
| SETR-MLA [6] | ViT-Large | $512 \times 512$ | 368.6 | 48.6 | 50.3 |
| MCIBI [33] | ViT-Large | $512 \times 512$ | >400 | - | 50.8 |
| Segmenter [8] | ViT-Large$^\dagger$ | $640 \times 640$ | 671.8 | 51.8 | 53.6 |
| StructToken [31] | ViT-Large$^\dagger$ | $640 \times 640$ | >700 | 52.8 | 54.2 |
| SegViT (*Shrunk*, ours) | ViT-Large$^\dagger$ | $640 \times 640$ | 373.5 | 53.9 | 55.1 |
| SegViT (ours) | ViT-Large$^\dagger$ | $640 \times 640$ | 637.9 | **54.6** | **55.2** |

55.2% in terms of mIoU with the ViT-Large backbone. It is 1.0% better than the recent StructToken [31] using the same backbone. Besides, our *Shrunk* version can also achieve a similar performance 55.1% with computational cost 373.5 GFLOPs which is much less than the ViT-Large backbone alone (612.3 GFLOPs).

**Results on COCO-Stuff-10K.** Table 2 shows the result on the COCO-Stuff-10K dataset. Our method achieves 50.3% which is higher than the previous state-to-the-art StrucToken by 1.2% with less computational cost. Our *Shrunk* version achieves 49.4% with 224.8 GFLOPs, which is similar to the computational cost of a dilated ResNet-101 backbone but with much higher performance.

**Table 2:** Experiment results on the COCO-Stuff-10K `test.` split. Following published methods, we report the results with multi-scale inference (denoted by 'ms'). The GFLOPs is measured at single scale inference with a crop size of $512 \times 512$.

| Method | Backbone | GFLOPs | mIoU (ms) |
|---|---|---|---|
| DANet [34] | Dilated-ResNet-101 | 289.3 | 39.7 |
| MaskFormer [15] | ResNet-101-fpn | 81.7 | 39.8 |
| EMANet [35] | Dilated-ResNet-101 | 247.4 | 39.9 |
| SpyGR [36] | ResNet-101-fpn | >80 | 39.9 |
| OCRNet [3] | HRNetV2-W48 | 167.9 | 40.5 |
| GINet [37] | JPU-ResNet-101 | >200 | 40.6 |
| RecoNet [38] | Dilated-ResNet-101 | >200 | 41.5 |
| ISNet [39] | Dilated-ResNeSt-101 | 228.3 | 42.1 |
| MCIBI [33] | ViT-Large | >380 | 44.9 |
| StructToken [31] | ViT-Large | >400 | 49.1 |
| SegViT (*Shrunk*, ours) | ViT-Large | 224.8 | 49.4 |
| SegViT (ours) | ViT-Large | 383.9 | **50.3** |

**Results on PASCAL-Context.** Table 3 shows the results on the PASCAL-Context dataset. We follow HRNet [40] to evaluate our method and report the results under 59 classes (without background) and 60 classes (with background). SegViT reaches mIoU 65.3% and 59.3% respectively for those two metrics that outperform the state-of-the-art methods using the ViT backbones with less computational cost.

**Table 3:** Expperiment results on the PASCAL-Context `val.` split. Following published methods, we report the results with multi-scale inference (denoted by 'ms'). $mIoU_{59}$: mIoU averaged over $59$ classes (without background). $mIoU_{60}$: mIoU averaged over 60 classes (59 classes plus background). Both metrics were used in the literature; and we report for the $60$ classes. The GFLOPs is measured at single scale inference with a crop size of $480 \times 480$.

| Method | Backbone | GFLOPs | $mIoU_{59}$ (ms) | $mIoU_{60}$ (ms) |
|---|---|---|---|---|
| RefineNet [41] | ResNet-152 | - | - | 47.3 |
| UNet++ [42] | ResNet-101 | - | 47.7 | - |
| PSPNet [11] | Dilated-ResNet-101 | 157.0 | 47.8 | - |
| Ding *et al.* [43] | ResNet-101 | - | 51.6 | - |
| EncNet [44] | Dilated-ResNet-101 | 192.1 | 52.6 | - |
| HRNet [40] | HRNetV2-W48 | 82.7 | 54.0 | 48.3 |
| NRD [45] | ResNet-101 | 42.9 | 54.1 | 49.0 |
| GFFNet [46] | Dilated-ResNet-101 | - | 54.3 | - |
| EfficientFCN [47] | ResNet-101 | 52.8 | 55.3 | - |
| OCRNet [3] | HRNetV2-W48 | 143.9 | 56.2 | - |
| SETR-MLA [6] | ViT-Large | 318.5 | - | 55.8 |
| Segmenter [8] | ViT-Large | 346.2 | - | 59.0 |
| SegViT (*Shrunk*, ours) | ViT-Large | 186.9 | 63.7 | 57.4 |
| SegViT (ours) | ViT-Large | 321.6 | **65.3** | **59.3** |

## 4.4 Ablation Study

In this section, we conduct the ablation study to show the effectiveness of our proposed methods.

**Effect of the ATM module.** Table 4 shows the effect of the ATM module. We set the SETR-naive as the baseline, which uses two $1 \times 1$ convolutions to get per-pixel classifications directly from the last layer of the ViT-Base transformer output. We can see that by applying the ATM module and supervise with a regular cross-entropy loss, ATM is capable of providing $0.5\%$ of performance boost. However, it is more beneficial to decouple the classification and mask prediction process and use the mask and classification supervision separately (3.1% increase).

**Ablation of using different layers as input for SegViT.** Table 5 shows the performance boost that multiple layers input can provide. We can see that the performance boost of feature maps from additional lower layers is obvious (+1.3%). We then involved more layers of features and see further performance gains. We empirically choose to use three layers for its best performance.

**Table 4:** Comparison between our proposed ATM module with other methods. 'CE loss' indicates the cross-entropy loss that is commonly used in semantic segmentation. The experiments are carried out on the ViT-Base backbone using ADE20K dataset.

| Decoder | Loss | mIoU (ss) |
|---|---|---|
| SETR | CE loss | 46.5 |
| ATM | CE loss | 47.0 (+0.5) |
| ATM | $\mathcal{L}_{mask}$ loss | **49.6 (+3.1)** |

**Table 5:** Ablation results of using different layer inputs to the SegViT structure on ADE20K dataset using ViT-Base as the backbone. Involving multi-layer features can bring obvious performance gain.

| | Used layers | mIoU (ss) |
|---|---|---|
| Single | [12] | 49.6 |
| Cascade | [6, 12] | 50.9 (+1.3) |
| Cascade | [6, 8, 12] | **51.3 (+1.7)** |
| Cascade | [3, 6, 9, 12] | 51.2 (+1.6) |

**Ablation for the ATM Decoder.** We conduct experiments to show the effectiveness of the proposed ATM decoder

**SegViT on hierarchical backbones.** Shown in Table 6, the SegViT structure is also able to apply to hierarchical backbones. We choose the most competitive methods Maskformer [15] and Mask2former [48] for comparison. Results indicate that even though our method is not designed for

**Table 6:** The experiments use the Swin-Tiny [18] backbone and are carried out on the ADE20K dataset. The GFLOPs are measured at single scale inference with a crop size of $512 \times 512$. QD: query-based down-saumping. QU: query-based upsampling.

| Method | mIoU (ss) | GFLOPs |
|---|---|---|
| Maskformer [15] | 46.7 | 57.3 |
| Mask2former [48] | **47.7** | 73.7 |
| SegViT (Ours) | 47.1 | **48.0** |

**Table 7:** Ablation of the QD module in terms of the targets and methods to down-sample. The experiments are carried out on the ViT-Large backbone of ADE20K dataset.

| Applied to | Methods | mIoU (ss) |
|---|---|---|
| Q | Conv | 44.5 |
| Q, K, V | Nearest | 52.6 |
| Q | Nearest | **53.9** |

hierarchical backbones, we can still achieve competitive performance while being efficient in terms of computational cost.

**Ablation for the QD module.** The motivation to use QD is to make use of the pre-train weights of the backbone. As in Table 7, if we use a stride 2 convolution with learnable parameters to down-sample the query, it will destroy the pre-train weights and dramatically decrease the performance. If the down-sampling is applied to both Q and (K, V), there will be an inevitable loss in information during the down-sampling process which is reflected in the weaker performance. We found that applying $2 \times 2$ nearest down-sampling on query only for the QD module is the better option.

**Ablation of the components in *Shrunk* structure.** Shown in Table 8, we studied the effect of each component (QD and QU) in the *Shrunk* structure. The results presented matches the structures illustrated in Fig. 3. When QD is applied, the performance decreases by 2.7% from the 'Single' ATM head. However, by applying QU, the performance is recovered. QD learns a non-linear down-sampling by the attention mechanism between key and query. One query will attend to several keys. QU is used to preserve the resolution and at the same time provide low-level feature information. We can see that by using QD and QU jointly, the performance can be retained and the computational cost is reduced. ATM module can also be used as the decoder to form our *Shrunk* structure to further boost performance.

**Table 8:** Ablation results of *Shrunk* version on the ADE20K dataset. The GFLOPs are measured at single scale inference with a crop size of $512 \times 512$ on ViT-Base backbone.

| Structure | QD | QU | Head | mIoU (ss) | GFLOPs |
|---|---|---|---|---|---|
| Single | | | SETR | 46.5 | 107.3 |
| Single | | | ATM | 49.6 (+3.1) | 115.8 |
| Naive *Shrunk* | ✓ | | ATM | 46.9 (+0.4) | 74.1 |
| *Shrunk* | ✓ | ✓ | ATM | **50.0 (+3.5)** | 97.1 |

## 5 Conclusion

We proposed an effective structure using plain ViT transformer backbones termed SegViT for the semantic segmentation task. For the first time, we utilize spatial information in attention maps for semantic segmentation. To implement this idea, we proposed an Attention-to-mask (ATM) module that can derive mask predictions during the attention calculation process. We show on a number of semantic segmentation benchmarks that our method is efficient and achieves state-of-the-art performance. We also proposed a *Shrunk* structure which is applied to the backbone and capable of reducing 40% of the computational cost while still maintaining competitive performance. We believe both structures can be strong paradigms, especially for semantic segmentation using ViT backbones. Last but not the least, our method still has some limitations. One of the limitations is that the large amount of GPU memory consumed by the global attention mechanism might not be supported by some devices, which might restrict the applicability of our structures.

**Acknowledgments** C. Shen's participation was in part supported by a major grant from Zhejiang Provincial Government. This work was also supported by the start-up funding of the University of Adelaide. [grant number 15130411]. This research was supported by Meituan.

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
