# OpenReview forum: "SegViT: Semantic Segmentation with Plain Vision Transformers"
_NeurIPS.cc/2022/Conference — NeurIPS 2022 Accept_

### Official Review · Reviewer_Nzhh · 2022-07-10

**Rating:** 5
**Confidence:** 4
**Soundness:** 2 fair
**Presentation:** 3 good
**Contribution:** 2 fair

**Summary:**

This paper present a semantic segmentation method based on the plain vision transformer (ViT). Specifically, it proposes the attention-to-mask (ATM) module to generate the pixel-level mask. In addition, to reduce the computational cost, it designs a query-based downsampling (QD) and upsampling module (QU) in the shrunk version. Experiments are conducted on three datasets and better results are obtained compared with previous methods.

**Questions:**

Please address the above concerns in the weakness.

**Limitations:**

The authors have addressed the limitations of the proposed method in large memory consumption.

**Strengths And Weaknesses:**

**Strength**

1. Exploring plain architecture for semantic segmentation is an interesting and promising direction. This paper make a forward step towards this direction.
2. The performance of SegVIT seems to be better than previous state-of-the-art methods.

**Weakness**

1. About the Attention-to-Mask module (ATM), it is implemented in cross-attention manner. But in fact, there is little difference with the standard classifier (a fc layer to map features to probability) for per-pixel classification in the normal semantic segmentation framework. Each learned token could be viewed as a classifier layer to map the pixel-level features into a probability with a Sigmoid function. In this sense, the ATM is similar to a standard classification layer.

2. About the Shrunk structure, I am confused about the query-based downsampling operation (QD). In line 172, it says to use the nearest sampling to reduce the token numbers. In this sense, it has nothing with the query based downsampling and is simply a standard downsampling operation. I am also confused about the implementation details on query-based upsampling operation (QU). It says to use a standard transformer decoder structure to upsample features. Is there any special design on the transformer decoder by incorporating the spatial information? More details are required on the decoder design.

3. About the two QU operations in the version (c) of Figure 3,  the downsampling ratio of lower QU is 1/16, and it is natural to think its output have smaller downsampling ratio like 1/8. However, from line 181-183, its output size seems to be 1/16, which is confused for me.

4. I think this paper should compare with the previous works PerceiverIO, which employs a similar downsampling-upsampling architecture for dense prediction with transformers. More discussion on the difference is required to better motivate the proposed method.

---

> ### Author Response · Authors · 2022-08-02
> **Response to Reviewer Nzhh**
>
> We thank the reviewer for the time and effort. We address the concerns
> proposed by Reviewer Nzhh in detail
>
> **1) About the Attention-to-Mask module (ATM), it is implemented in cross-attention manner. But in fact, there is little difference with the standard classifier (a fc layer to map features to probability) for per-pixel classification in the normal semantic segmentation framework. Each learned token could be viewed as a classifier layer to map the pixel-level features into a probability with a Sigmoid function. In this sense, the ATM is similar to a standard classification layer.**
>
> *A: If only considering this layer, by simply replacing the attention mask with a per-pixel output, the performance will drop by 0.6\%. Moreover, we do not claim the ATM module for our sole contribution in this paper. We use ATM to propose a structure that is capable of solving semantic segmentation tasks with a lightweight structure and high performance, which is non-trivial. Also, we dig into the topic of how to make use of the existing pre-trained weights while greatly reducing the computational cost. The Shrink structure we proposed is capable of reducing 40\% computational cost without damaging the pre-trained weight.*
>
> **2) About the Shrunk structure, I am confused about the query-based downsampling operation (QD). In line 172, it says to use the nearest sampling to reduce the token numbers. In this sense, it has nothing with the query based downsampling and is simply a standard downsampling operation.**
>
> *A: For the QD, our approach is to reduce the size of the query tokens while keeping the resolution of the keys and values.
> When passing through a transformer layer, the output is the weighted sum of the 'value (feature map)' according to the attention map between 'query' tokens and the 'key' tokens. This is non-linear downsampling which will pay more attention to the important regions. While a standard downsampling operation will reduce the spatial size of the 'feature map' by a convolution layer with stride = 2, this is a grid sampling that reduces the feature map regularly. We want to directly use the existing pre-trained weights of plain backbones to enable the extension to other tasks.  In table 7 we demonstrated how QD compares to the simple nearest downsample over qkv (q vs qkv: 52.6 vs 53.9)*
>
> **3) I am also confused about the implementation details on query-based upsampling operation (QU). It says to use a standard transformer decoder structure to upsample features. Is there any special design on the transformer decoder by incorporating the spatial information? More details are required on the decoder design.**
>
> *A: The first QU is made with random initialized 1/16 resolution tokens as queries and the 1/16 resolution layer outputs of ViT as keys and values. This QU records the information before the QD is applied to the ViT backbone. The structure is a typical transformer decoder, with a self-attention layer and a cross-attention layer. The second QU is the previous QU's output as queries and the downsampled ViT output as keys and values. Even though the keys and values have a smaller resolution, the output of a transformer decoder is dependent on the queries. Thus the output of the QU is still 1/16 in resolution.*
>
> **4) About the two QU operations in the version \(c\) of Figure 3, the downsampling ratio of lower QU is 1/16, and it is natural to think its output have smaller downsampling ratio like 1/8. However, from line 181-183, its output size seems to be 1/16, which is confused for me.**
>
> *A: We employ the QD and QU to reduce the computational cost in version \(c\) of Figure 3. The length of the q determines the spatial size of the output. We first apply a smaller q (QD) to reduce the computational cost of the backbone (from 1/16 to 1/32), then we apply a larger q (QU) to upsampling the final feature map. With this simple structure, we can reduce the computational cost while maintaining performance. The accuracy will be significantly reduced if this QU and QD are replaced with a standard convolution up-sampling and down-sampling.*

---

> > ### Comment · Reviewer_Nzhh · 2022-08-09
> > **Response to rebuttal**
> >
> > Thanks for the detailed rebuttal. It addressed partial of my concern.
> >
> >  I still have concern on the novelty of ATM (also pointed by reviewer Eyo8). Although the rebuttal provides the result of per-pixel output (drop by 0.6%), I really fail to get the essential difference between the proposed ATM and the standard classifer (fc layer). Also, the reviewer Eyo8 points out the similarity of ATM with MaskFormer. So I keep my rating as borderline reject.

---

> > > ### Author Response · Authors · 2022-08-09
> > > **The novelty of this work**
> > >
> > > Dear all,
> > >
> > > The novelty of this work is **not limited** to the design of the ATM.  It explores a powerful and efficient decoder structure based on the ViT backbone. ATM is an implementation of decoupling the mask and the class to enable the new loss. It is simpler and more straightforward than the design in Maskformer (also more powerful on top of the ViT backbone).
> > >
> > > SegVit is an elegant, concise and efficient solution, resulting from a design that is more suitable for the transformer structure.
> > > Although we all know and admit that
> > >
> > > *1. The fc layer cannot decouple mask and classification, leading to low performance.*
> > >
> > > *2. The design of the MaskFormer is very complex (with Hungarian matching), the calculation is heavy (two different branches to decouple), and the effect is not good on ViT (4% decreased in mIoU with two times the computational cost).*
> > >
> > > Do we still insist on rejecting this solution because of the similarity of the implementation for one small block?
> > >
> > > Consider if now we want to use CLIP or DINO, a ViT-based backbone that is pre-trained on the large-scale dataset, the SegViT will be a good choice. The class tokens can be replaced with language embeddings. It will converge faster and generalize better. Moreover, the Shrunk structure can reduce the computational cost in these pre-trained models without damaging the pre-trained knowledge. Previous work based on transformers can not achieve this! They all require re-training.
> > >
> > > We sincerely hope that the reviewers can reconsider the simplicity and advantages of our work The exploration of Vit backbone is our innovation.
> > >
> > > Best wishes,
> > >
> > > Authors

---

### Official Review · Reviewer_Eyo8 · 2022-07-11

**Rating:** 4
**Confidence:** 5
**Soundness:** 4 excellent
**Presentation:** 4 excellent
**Contribution:** 2 fair

**Summary:**

The paper proposed a plain-ViT based semantic segmentation framework, which uses an Attention-to-Mask decoder to aggregate image features and a Shrunk structure to save computational cost.

**Questions:**

1. Considering the similarity of MaskFormer and SegViT, I am curious about the result of MaskFormer + ViT + multi-layer-feature (use the same three ViT layer inputs to MaskFormer transformer decoder). I think the result may be similar with SegViT.
2. In the paper, it claims that the SegViT framework has a quite good performance with Swin Transformer backbone and lists the results with Swin-Tiny. I am curious about SegViT + Swin-Large result and its comparison with MaskFormer and Mask2Former.


**Ethics Review Area:**

["I don’t know"]

**Limitations:**

The authors have addressed the limitations and potential negative societal impacts.

**Strengths And Weaknesses:**

Strengths:
1. The paper proposed SegViT framework and achieved a SOTA performance based on a plain ViT backbone.
2. The paper is well written and clear to understand.

Weaknesses:
1. I doubt the novelty of the design of ATM module. Since the MaskFormer framework has been proposed for over half a year, the ATM module is similar to the MaskFormer transformer decoder module. The only difference is that the mask output of MaskFormer is generated by the multiplication of the final output query tokens and image features, while the mask output of ATM is from the multiplication of an intermediate variable K inside the transformer layer and the image features. The difference is not obvious. The SegViT framework is just like MaskFormer + ViT + multi-layer-feature.
2. Table 4 shows that ATM has a relatively low performance gain of about 0.5% to SETR. It shows that the performance of ATM is even worse than Segmenter (Since the result of Segmenter is 0.8% better than SETR in Talbe 1)?
3. Also, in table 4, it shows that by using L_mask loss the mIoU result increases about 2.6% than using CE loss only. However, Table 1 shows that the result of SegViT is about 2.3% better than Segmenter baseline (which only uses CE loss). If it shows that the performance gain is all from the new loss design but not from the framework architecture.

---

> ### Author Response · Authors · 2022-08-02
> **Response to Reviewer Eyo8**
>
> We thank the reviewer for the time and effort. We address the concerns proposed by Reviewer Eyo8 in detail.
>
> **1) I doubt the novelty of the design of ATM module. Since the MaskFormer framework has been proposed for over half a year, the ATM module is similar to the MaskFormer...**
>
> *A: We made ablation studies on transferring maskformer and mask2former to ViT backbones. For maskformer, we adopt our code base using mmsegmentation using the exact same hyper-parameters that we use, the performance on ViT-base is (46.7 maskformer vs 51.2 Ours). For mask2former, we use the detectron2 code base and inherit all the hyper-parameters that mask2former original code use just changes the backbone to Vit-base, the performance is 48.3.
> The design of this paper is based on the non-hierarchical ViT backbone to solve the semantic segmentation problem. Our conceptual difference lies in the assumption that we assign each class to one fixed class token and generate the mask of the specific class directly. While maskformer depends on Hungarian matching and each learnable query corresponds to the spatial information rather than category information. With this difference in assumption, our ATM module do not need to add positional embedding like maskformer.
> Also, our major contribution claimed is to develop a lightweight framework that is efficient and effective on plain backbones. Experiments show that our method achieved SOTA performance using the least amount of computational cost among all existing plain backbone methods.  Please refer to the response to reviewer YDPR for more details.*
>
> **2) Table 4 shows that ATM has a relatively low-performance gain of about 0.5\% to SETR. It shows that the performance of ATM is even worse than Segmenter (Since the result of Segmenter is 0.8\% better than SETR in Talbe 1)?**
>
> *A: Table 4 is an ablation study to demonstrate the effect of different losses on ATM structure. We utilize the ATM module to propose a structure that is capable of being lightweight and effective. The contribution of the paper is not limited to the design of ATM. How to apply the ATM in the cascade and shrunk structure is also a non-trivial work. In Table 1 we show that our proposed structure has higher performance than Segmenter while having lower computational cost.*
>
> **3) Also, in table 4, it shows that by using $L_{mask}$ loss the mIoU result increases about 2.6\% than using CE loss only. However, Table 1 shows that the result of SegViT is about 2.3\% better than Segmenter baseline (which only uses CE loss). If it shows that the performance gain is all from the new loss design but not from the framework architecture.**
>
> *A: We have a class token from the output of the transformer block and a segmentation mask from the attention map. These two outputs can be merged into one logits map to enable training with CE loss. But if the network can only generate a pixel level logits map, like in SETR and Segmentor, the mask loss can not be successfully applied. The results in Table 4 demonstrate that supervising the attention masks and the class tokens separately is a more natural way. This supervision is enabled by the ATM module in a simple and elegant way.*
>
>
> **4) Considering the similarity of MaskFormer and SegViT, I am curious about the result of MaskFormer + ViT + multi-layer-feature (use the same three ViT layer inputs to MaskFormer transformer decoder). I think the result may be similar with SegViT.**
>
> *A: We made ablation studies on transferring maskformer and mask2former to ViT backbones. For maskformer, we adopt our code base using mmsegmentation using the exact same hyper-parameters that we use, the performance on ViT-base is (46.7 maskformer vs 51.2 Ours).
> For mask2former, we use the detectron2 code base and inherit all the hyper-parameters that mask2former original code use just changes the backbone to Vit-base, the performance is 48.3. Please refer to the response to reviewer YDPR for more details.
> The low performance is possible due to 1, the use of Hungarian matching that brings slow convergence which is unnecessary for semantic segmentation, and 2, the use of FPN which introduced unnecessary parameters.*
>
> **5) In the paper, it claims that the SegViT framework has a quite good performance with Swin Transformer backbone and lists the results with Swin-Tiny. I am curious about SegViT + Swin-Large result and its comparison with MaskFormer and Mask2Former.**
>
> *A: The performance for the proposed structure and the MaskFormer are similar on Swin-Large but has a large performance gap on ViT backbones. That is because we design our structure for non-hierarchical ViT backbone with a large feature map resolution. The cascade and shrunk structures are all sepcifically designed for ViT structures. It is meaningful to explore the potential of the plain backbone. For example, the plain backbone is widely used in large pre-trained tasks. Our structure can be easily combined with CLIP or other mutimodel learning.*

---

### Official Review · Reviewer_oFbW · 2022-07-13

**Rating:** 5
**Confidence:** 3
**Soundness:** 3 good
**Presentation:** 3 good
**Contribution:** 2 fair

**Summary:**

The authors deal with ViT-based semantic segmentation. In particular, they use a set of learnable tokens (each corresponding to a semantic class) which decode the outputs of the ViT-based backbone into per-class semantic masks. This is accomplished by multiple layers of cross attention between class token and ViT tokens. Rather than use a dot product like mechanism to produce similarity between a class token and spatial features, they directly supervise the cross attention maps using a sigmoidal output. Furthermore, they introduce a down/upsampling technique to mimic the general idea of an efficient multi-scale prediction head. Their results are quite good even when compared to some of the best recent models and their QD module provides some computation/performance tradeoffs.

**Questions:**

1. I'm slightly confused about why one needs class predictions on the output tokens? Is this because the attention mechanism is bad at producing low confidence values when the class does not exist? Do you just multiply this probability by the sigmoid output?
2. Table 4 could be more convincing. The loss formulation is not novel, so why not also apply it to SETR?
3. It seems reasonable to provide a "dot product"-based baseline (similar to the decoder in Segmenter) to tease where attention (which itself is a dot product to some degree in this scenario) actually makes a difference.
4. Does this work when just supervising the aggregated output (a single per class segmentation loss)?

**Limitations:**

I believe so.

**Strengths And Weaknesses:**

Strengths:

1. This is a well written paper and the approach is quite clean
2. The results presented are quite good as well, achieving at/near SOTA against competitive models

Weaknesses

1. The idea is still related to the idea of dot product based segmentation (from some class embedding). I think a good deal of experiments might need to be performed to actually understand the technical contribution.
2. While the results are good, related work like Segmenter is not far off from the performance presented here and shares some significant similarities with this method.

---

> ### Author Response · Authors · 2022-08-02
> **Response to Reviewer oFbW**
>
> We thank the reviewer for the time and effort. We address the concerns proposed by Reviewer oFbW in detail.
>
> **1) While the results are good, related work like Segmenter is not far off from the performance presented here and shares some significant similarities with this method.**
>
> *A: We aim at proposing efficient segmentation networks with a plain backbone. Segmenter employs the class tokens as the input of the ViT backbone, which we found it unnecessary and low efficient. By introducing the class tokens as the input of the ATM, and supervising the attention map, our computational cost is much smaller than Segmenter (Gflops w/o. backbones: ours 25 vs Segmenter 59) while we outperform Segmenter by 1.4\% on  ADE20k dataset.*
>
> **2) I'm slightly confused about why one needs class predictions on the output tokens?
> Is this because the attention mechanism is bad at producing low confidence values when the class does not exist? Do you just multiply this probability by the sigmoid output?**
>
> *A: Yes, not only the attention mechanism, the mask loss which is originally used by DETR is not good at producing low confidence values and requires the class predictions to indicate whether the specific class exists in the input image or not. This probability is just multiplied by the mask which is the sigmoid output of the attention.*
>
> **3) Table 4 could be more convincing. The loss formulation is not novel, so why not also apply it to SETR?**
>
> *A: As in the question above, the mask loss is the combination of dice loss and binary cross-entropy loss for every single mask that has the ground truth. If there is not a class prediction to indicate, the performance can be rather low (37.4 applied to SETR, SETR 46.5 for regular CE loss)*
>
> **4) The idea is still related to the idea of dot product-based segmentation (from some class embedding). I think a good deal of experiments might need to be performed to actually understand the technical contribution.
> It seems reasonable to provide a "dot product"-based baseline (similar to the decoder in Segmenter) to tease where attention (which itself is a dot product to some degree in this scenario) actually makes a difference.**
>
> *A: We thank the reviewer for the great suggestion. We run a variation of our work by simply changing the output from the attention mask to the dot product. The result decreased from 51.2 to 50.6 on Vit Base. This demonstrates that the attention output can improve the final results. Moreover, we observe with the ATM, the network can converge faster compared to the "dot product" baseline.*
>
> **5) Does this work when just supervising the aggregated output (a single per class segmentation loss)?**
>
> *A: No it won't work. The classification logits are needed to indicate the existence of its class. In the dataset, there can be a lot of similar categories e.g. house VS building, lake VS river VS sea. They can all have similar masks. The classification logits provide indications of which class to take.*

---

> > ### Comment · Reviewer_oFbW · 2022-08-10
> > **Response to rebuttal**
> >
> > Thanks for the responses. I hope some of these comparisons could make it into a future version. I'm generally still quite okay with this submission living at the borderline. My opinion would be closer to accept than reject, but as others reviewers have mentioned, there is something left to be desired with respect to novelty.

---

> > > ### Author Response · Authors · 2022-08-10
> > > **Thanks for the support!**
> > >
> > > We thank the reviewer for the support!
> > >
> > > Regarding the novelty, please refer to the feedback from reviewer Nzhh. Our initial goal is to develop a simple and efficient structure for existing plain ViT backbones.  The ATM model decoupled the mask and the class prediction on one transformer block. The cascade structure merges multi-level features with the help of attention machines. The Shrunk structure proposed a method that can reduce half of the computational cost in the backbone without damaging the pre-trained weight. All these contributions are developed purely based on ViT structure.
> > >
> > > The plain backbone also has potential in many areas. It is suitable to merge with multimodality data, which a heuristical backbone can not do. If this direction is successful, it will enable the use of original ViT backbones for semantic segmentation; this will decouple the pre-training design from the fine-tuning demands, maintaining the independence of upstream vs. downstream tasks.
> > >
> > > Thanks again.

---

### Official Review · Reviewer_YDPR · 2022-07-18

**Rating:** 5
**Confidence:** 4
**Soundness:** 2 fair
**Presentation:** 3 good
**Contribution:** 2 fair

**Summary:**

The paper introduces SegViT, a semantic segmentation framework with plain ViTs as backbones. One of the core technical contributions is the proposed Attention-to-Mask (ATM) block, which generating masks from the intermediate attention maps between class embeddings and key maps. In addition, a shrunk structure is then proposed to save computational cost while maintaining the performance. Based on plain ViT networks only, SegViT obtains state-of-the-art results on three semantic segmentation datasets (ADE20K, PASCAL-Context and COCO-Stuff-10K).

**Questions:**

SegViT still utilizes FPN-like architecture to fuse low-level and high-level features from the backbone. However, some previous works [*1, *2, *3] challenge the essential of such feature fusion. It will be interesting if the authors show experiments where ATM module only relates to the last layer of ViT encoder. For example, like the experiments in Table 5, the authors may try a cascade design while utilizing the last layer multiple times, or just follow ViTDet fashion [*3].

[*1] Zhang et al. ExFuse: Enhancing Feature Fusion for Semantic Segmentation. ECCV 2018.

[*2] Chen et al. You Only Look One-level Feature. CVPR 2021.

[*3] Li et al. Exploring Plain Vision Transformer Backbones for Object Detection. Tech report.


**Limitations:**

Limitations are mentioned in the conclusion. Although I think more discussion and comparisons with [15] are required in the paper.

**Strengths And Weaknesses:**

Pros

1. The paper is well motivated. Recently many works (e.g. [*1]) have realized even plain ViTs could have rich representation capacity, which however requires special optimization (e.g. masked image modeling) or other architectural modifications for downstream tasks. I am pleased that the paper demonstrates that plain ViTs can obtain as good results as the hierarchical counterparts (e.g. [15, 47]) on segmentation tasks, which may encourage simpler and unified network design principles.

2. Strong results are reported in the paper. For example, on ADE20K val a model with ViT-L backbone achieves 55.2 mIoU, which is very competitive even among more sophisticated networks, such as Swin-L and MViT.

3. The motivation of ATM module sounds reasonable to some extent: intuitively a good attention mask should cover the foreground of the given object (or class). Therefore, it is possible to generate mask directly from the attention matrix.

Cons

1. My major concern is that the technical novelty is relatively limited. The overall framework is very similar to MaskFormer [15] and Mask2Former [47]. Compared with [15], the major difference on the technical details is, [15] generate masks from the product of the mask embedding and the per-pixel embedding, while in the paper the mask is directly derived from the attention weights. However, I do not think it differs much. Although [15, 47] mainly evaluate on hierarchical backbones, theoretically they can also be equipped with plain networks. In addition, the proposed QU/QD layers are not novel (also sounds irrelevant to the main topic of the paper), since many previous works, e.g. PVT [17], also adopt similar blocks to reduce computational cost. In conclusion, I think the contributions claimed in the introduction seems not significant.

2. According to Table 4 and Line 234-239, in the proposed ATM block, the separated supervision of classification and mask prediction is the most important design principle. However, it is not originally proposed in the paper as [15, 16] already introduces the paradigm. It further weakens the significance of the proposed method.

[*1] Li et al. Exploring Plain Vision Transformer Backbones for Object Detection. Tech report.

---

> ### Author Response · Authors · 2022-08-02
> **Response to Reviewer YDPR**
>
> **1.1) The overall framework is very similar to MaskFormer [15] and Mask2Former [47] ...**
>
> *A: We thank the reviewer for the time and effort. We address the concerns proposed by Reviewer YDPR in detail.
> The design of this paper is based on the non-hierarchical ViT backbone to solve semantic segmentation problem. The main difference with the MaskFormer are summarized as follows:*
> 1. *Our conceptual difference lies in the assumption that we assign each class to one fixed class token and generate the mask of the specific class directly. While maskformer depends on Hungarian matching and each learnable queries corresponds to the spatial information rather than category information.*
> 2. *Our ATM module do not need to add positional embedding like MaskFormer, because we make use of the attention map between the class token and the feature map.*
> 3. *Maskformer is highly dependent on the typical CNN structures, for example, FPN is used to merge the features from hierarchical backbones. However, in our work, we enhance the representation ability of the class tokens by transformer blocks. The masks are generated by the sum of the multi-layer attention maps.*
>
> *Also, our major contribution claimed is to develop a lightweight framework on plain backbones. Plain backbones have a lot of potential applications, for example, linking the text with the images (CLIP), unsupervised learning with larger scale datasets and so on. We are making use of the inartistic features of the vision transformers. Experiments show that our method achieved SOTA performance using the least amount of computational cost among all existing plain backbone methods.*
>
> *Besides, to further reduce the computational cost that the Vit backbone brings, we introduced the Shrunk structure, which is able to reduce computational cost on the backbone while still using the original pre-train weights. In table 7, we showed that for ViT backbones, regular methods like using a convolution layer to downsample the feature map, can seriously decrease the performance and make the pre-trained weights no longer effective. While using our Shrink structure, the performance can still be maintained. This enables the extension of the larger pre-trained models, e.g., CLIP, DINO, instead of training a new one from scratch.*
>
> **1.2) Although [15, 47] mainly evaluate on hierarchical backbones, theoretically they can also be equipped with plain networks.**
>
> *A: Transferring the decoupled output type of [15, 47] is not a **trivial work**. A straightforward transfer can not obtain competitive performance. We made ablation studies on transferring MaskFormer and Mask2Former to ViT backbones. The results are shown in the following table:*
>
> | Method | Backbone| mIoU on ADE20k|GFLOPs wo Backbone|
> | :---- | :----: |:----: |:----: |
> |MaskFormer| Vit-base|46.7 | 65|
> |Mask2Former |Vit-base| 48.3 | 43|
> |Ours | Vit-base| 51.2 | 32|
>
> **1.3) In addition, the proposed QU/QD layers are not novel ...**
>
> *A: For the QD, our approach is to reduce the size of the query tokens while keep the resolution of the keys and values.*
> *When passing through a transformer layer, the values are weighted and summed by the attention map between query tokens and the key tokens. This is non-linear downsampling that will pay more attention to the important regions. While in PVT, they reduced the spatial size of keys and values by a convolution layer with strides, this is a grid sampling that reduces the feature map regularly. Moreover, PVT focuses on designing a new hierarchical backbone and trains the backbone from the scratch.  While we want to directly use the existing pretrain weights of plain backbones to enable the extension to other tasks. Also, in table 7 we demonstrated how additional convolution with strides downsampling will seriously reduce the performance and how QD compares to the simple nearest downsample over qkv (q vs qkv, 52.6 vs 53.9).*
>
> **2) According to Table 4 and Line 234-239, in the proposed ATM block, the separated...**
>
> *A: ATM is a way to enable the network to be supervised with the new loss. Our final mask (attention map) is generated related to every single semantic class. It is not a per-pixel representation. Thus, supervising with CE can not achieve a good performance.
> Our main contribution is to discuss and propose a lightweight and effective structure for plain backbones like ViT. It is a simple and elegant solution with competitive results on ViT backbones.
> Directly applying the MaskFormer decoder can not achieve competitive performance.*

---

> > ### Author Response · Authors · 2022-08-02
> > **part 2**
> >
> > **3) SegViT still utilizes FPN-like architecture to fuse low-level and high-level features ...**
> >
> > *A: The architecture, even though takes in features from multiple layers as input, is not in the purpose of FPN, of which the intention is to merge the features from multiple layers and generate a stronger feature map. In our structure, every layer is directly applied to the ATM and then the generated masks are summed. There is no feature maps merge process. Also, in the experiment, we found the information that the lower level provides are 'refining' the output (with most areas zero and only having value on edge areas), rather than 'merge'.
> > We also made ablation studies on expanding the resolution of the last layer as the VitDet. There was no obvious improvement. It's can partially because the semantic segmentation task is not as sensitive to multi-scale as the detection task. Also, without larger-scale feature maps (1/16 for all in ViT) a deconvolution or bilinear upsampling is not able to provide more semantic information.*

---

> > > ### Comment · Reviewer_YDPR · 2022-08-08
> > > **Post-rebuttal comments**
> > >
> > >
> > > Thanks for the very detailed rebuttal. But I still have a few concerns on the novelty. Here are some comments:
> > >
> > > ### About the differences to MaskFormer. I am not agree with the feedback:
> > >
> > > * First, the rebuttal says "MaskFormer depends on Hungarian matching and each learnable queries corresponds to the spatial information rather than category information", but actually MaskFormer mentions fixed matching in Sec 3.2.
> > >
> > > * Second, the rebuttal claims "ATM module do not need to add positional embedding like MaskFormer". But to my knowledge, MaskFormer requires positional embedding for instance segmentation, otherwise the learned mask embeddings cannot distinct instances of the same class but in different position. However, for semantic segmentation, it should not be necessary.
> > >
> > > * Third, the rebuttal introduces some advantages of plain ViTs, e.g. linking the text with the images (CLIP), unsupervised learning with larger scale datasets and so on. However, to my knowledge there is no evidence that hierarchical ViTs have any drawbacks on those tasks.
> > >
> > > As a conclusion, I suggest the authors may consider carefully on justifying the novelty over MaskFormer and Mask2Former in the revised version, no matter whether the paper is accepted immediately.
> > >
> > > ### About transferring MaskFormer and Mask2Former to plain ViTs
> > >
> > > The experiment is very interesting. I recommend the authors add the results into the manuscript with proper explanation. I am not very clear why those methods cannot generalize well on plain ViTs. Is there any insights behind the implementation?
> > >
> > > ### About QU/QD
> > >
> > > I agree with the justification on PVT. It is also recommended to cite [*1, *2], as those works also involve downsample/upsample with queries.
> > >
> > > [*1] Ma, Xuezhe, et al. "Luna: Linear unified nested attention." NeurIPS 2021.
> > >
> > > [*2] Ryoo, Michael, et al. "Tokenlearner: Adaptive space-time tokenization for videos." NeurIPS 2021.
> > >
> > > ### About Table 4 and Line 234-239
> > >
> > > I am confused with the rebuttal. To my understanding, MaskFormer does not require per-pixel classification either (for each mask embedding, MaskFormer only generates a binary mask, which is very similar to this paper). So, my concern on Table 4 still exists: it seems the most performance gain comes from $\mathcal{L}_{mask}$, however, which I think cannot be regarded as the **unique** contribution of this paper.

---

> > > > ### Author Response · Authors · 2022-08-10
> > > > **About the differences to MaskFormer**
> > > >
> > > > Thanks for the reviewer's reply!
> > > > The difference with Maskformer is not our novelty. The motivations of these two papers are totally different.
> > > >
> > > > The MaskFormer aims at proposing a unified network structure that can handle all segmentation tasks together. The main contribution was the decoupled mask and segmentation loss. We respect and appreciate their contributions. Our main goal is to explore a simple and efficient network structure based on plain backbone ViT.  Imagine if researchers in another field now want to introduce a semantically segmented head on the structure of a ViT, they may not have the same granular knowledge of the details of two papers (MaskFormer and Ours) as you do. Without our work, their direct transfer MaskFormer would have degraded performance and slow convergence.
> > > >
> > > > Regarding your detailed concern, we would like to provide more information as following:
> > > >
> > > > **First, the rebuttal says "MaskFormer depends on Hungarian matching and each learnable queries corresponds to the spatial information rather than category information", but actually MaskFormer mentions fixed matching in Sec 3.2.**
> > > >
> > > > *The fixed matching in Sec 3.2. in MaskFormer generates the learnable tokens from the backbone feature and generates the mask from the FPN features. They found that the semantic segmentation branch is possible to gain a 0.5\% performance boost due to Hungarian matching. However, we found when we learn the masks from the attention map and generate the class tokens within one attention block, removing Hungarian matching will not lead to a performance drop.*
> > > >
> > > > **Second, the rebuttal claims "ATM module do not need to add positional embedding like MaskFormer". But to my knowledge, MaskFormer requires positional embedding for instance segmentation, otherwise the learned mask embeddings cannot distinct instances of the same class but in different position. However, for semantic segmentation, it should not be necessary.**
> > > > *Yes. We agree with your opinion. That is why we removed the positional embedding. This is the difference, not the novelty over Maskformer.*
> > > >
> > > > **Third, the rebuttal introduces some advantages of plain ViTs, e.g. linking the text with the images (CLIP), unsupervised learning with larger scale datasets and so on. However, to my knowledge there is no evidence that hierarchical ViTs have any drawbacks on those tasks.**
> > > >
> > > > *Hierarchical ViTs may or may not have drawbacks. However, it requires a lot of computational resources to re-training the hierarchical ViTs on those large datasets, following DINO and CLIP. Our research enables the original ViT to do semantic segmentation efficiently. This will decouple the pre-training design from the fine-tuning demands, maintaining the independence of upstream vs. downstream tasks. Moreover, the original ViT does not have local convolutional layers, which enable the application of tricks from NLP, such as prompt tuning.*
> > > >
> > > > **About transferring MaskFormer and Mask2Former to plain ViTs.**
> > > >
> > > > *Thanks for your encouragement! We will add this result in our revision and hope we will have the chance to share these findings with the research community. We made fair implementations of maskformer to Vit backbones and used both our set of hyper-parameters and mask2former's own hyper-parameters. The possible reasons are in details but not trivial:*
> > > >
> > > > **1. The FPN in MaskFormer and the Cascade structure in our SegVit.**
> > > >
> > > > *For ViT backbones, the most semantic information-rich feature maps are the last layer. The intention of the maskformer structure in terms of semantic segmentation is to make good use of the multi-level information. From high-level small resolution to gather category/locational information and from low-level merged features to get mask details. However, this is not necessary for ViT backbones. We did experiments with no additional encoders, no fpn 3x3 convs, and fixed matching which is now very close to our ATM structure, the performance is (50.6 vs 51.2(ours)) *
> > > >
> > > > **2. Hungarain matching and our fixed class tokens.**
> > > >
> > > > *Another reason is the Hungarain matching involved. We found that with the involve of the matching, the converge speed is dramatically slowed on ViT backbones.*
> > > >
> > > > This implies that the shrift to ViT backbone from maskformer is a non-trivial task.

---

> > > > ### Author Response · Authors · 2022-08-10
> > > > **Other feedbacks**
> > > >
> > > > **About QU/QD**
> > > >
> > > > *We thank YDPR for the great advice, we will add those cites to the paper.*
> > > >
> > > > **About Table 4 and Line 234-239**
> > > > *In Maskformer, they first the queries to have multiple transformer decoder calculation with the high-level feature maps from the backbone. The results are 100 queries. Then they use two FC layers to generate the classification logits and the 1x1 kernels to have 1x1 convolutions with the low-level features. Thus for maskformer to process, it requires a high-level feature map and a low-level feature map. (They can use purely on high-level feature map only, however, the hierarchical backbone has 1/32 resolution backbones which is too small for segmentation, and with multiple steps of downsampling, the detailed information is lost.)
> > > > Our ATM managed to make use of the attention which is a by-production along the calculation as the source to generate the mask, which enables the mask loss to be carried on any single feature map. This is a simple implementation, however, we proved its effectiveness and find a good application to use it (since on hierarchical backbones the single stage feature map information is not rich enough, ATM is not very effective).
> > > > For Table 4, the conclusion can only draw that compared with setr, we enabled the use of mask loss via the ATM module and boost the performance by 3.1\%. That is the effect of this module only.
> > > > With this module, we proposed the cascaded structure and the shrink structure, which, without atm is not possible.*

---

### Author Response · Authors · 2022-08-07
**Have we addressed your concerns?**

Dear all reviewers,

Do our responses answer your questions? Please let us know if you have any more questions. Thank you for your time.

Best wishes,
All the authors

---

### Meta-Review · Area_Chair_zxgL · 2022-08-26

**Recommendation:** Accept
**Confidence:** Certain

**Metareview:**

This submission has received comments from 4 official reviewers. The authors have made very detailed replies to the reviewer's comments. The authors and reviewers had quite rich discussions. After these discussions, 3 reviewers recommended weak acceptance, and 1 recommended rejection.

For the novelty concerns, the authors clarify them during the rebuttal. The reviewers have also recommended comparing with recent semantic segmentation methods using ViTs. Missing comparisons should be included in the final version, including comparisons with

[1] Ma, Xuezhe, et al. "Luna: Linear unified nested attention." NeurIPS 2021.
[2] Ryoo, Michael, et al. "Tokenlearner: Adaptive space-time tokenization for videos." NeurIPS 2021.
[3] Wu, Yu-Huan, et al. "P2T: Pyramid Pooling Transformer for Scene Understanding", IEEE TPAMI, 2022.


Only reviewer Eyo8 recommends borderline rejection. The authors have made quite a detailed rebuttal but we have not heard from the reviewer after the rebuttal.

Thus, the AC would like to recommend acceptance.


**Award:**

No

---

### Decision · Program_Chairs · 2022-09-14

Accept